# Metabolic Syndrome Prevalence among High School First-Year Students: A Cross-Sectional Study in Taiwan

**DOI:** 10.3390/nu14173626

**Published:** 2022-09-02

**Authors:** Chin-Yu Ho, Kuan-Yu Fan, Ernest Wen-Ruey Yu, Ting-Fang Chiu, Chi-Hua Chung, Jason Jiunshiou Lee

**Affiliations:** 1Department of Family Medicine, Taipei City Hospital Yangming Branch, Taipei 111, Taiwan; 2General Education Center, University of Taipei, Taipei 100, Taiwan; 3Department of Psychology, Soochow University, Taipei 111, Taiwan; 4School of Medicine, National Yang Ming Chiao Tung University, Taipei 112, Taiwan; 5Department of Pediatrics, Taipei City Hospital Zhongxiao Branch, Taipei 115, Taiwan; 6Institute of Public Health, National Yang Ming Chiao Tung University, Taipei 112, Taiwan; 7Department of Health Care Management, National Taipei University of Nursing and Health Sciences, Taipei 112, Taiwan

**Keywords:** adolescent metabolic syndrome, cardiovascular diseases, senior high school, vocational high school, physical education curriculum, health education program

## Abstract

Different types of high schools in Taiwan have the same physical education curriculum. In this cross-sectional study, we investigated the difference in the prevalence of metabolic syndrome between senior and vocational high school students. We retrospectively collected health check-up data from 81,076 first-year senior and 68,863 vocational high school students in Taipei City from 2011 to 2014, including their blood pressure, height, weight, waist circumference, fasting blood glucose, total cholesterol, triglyceride, and HDL-c levels. The prevalence of metabolic syndrome was determined using definitions from the Taiwan Pediatric Association (TPA), International Diabetes Federation (IDF), and de Ferranti et al. The prevalence of metabolic syndrome was 1.73% (senior and vocational high school students: 1.22% and 2.33%, respectively) using TPA criteria, 1.02% (0.69% and 1.40%, respectively) using IDF criteria, and 5.11% (3.92% and 6.51%, respectively) using de Ferranti et al. criteria. The most prevalent risk factors overall were increased blood pressure and central obesity. Given the significantly higher prevalence of metabolic syndrome in vocational school students regardless of the criteria, and that metabolic syndrome causes future adult health risks, the physical education curriculum and health education program in vocational schools should be strengthened to decrease the risk and prevalence of metabolic syndrome.

## 1. Introduction

Cardiovascular diseases (CVDs) are the leading non-communicable diseases in developed and developing countries and the leading cause of death worldwide. The prevalence of CVDs, excluding hypertension, was 9.0% and 9.2% in the United States in 2016 and Europe in 2014, respectively, and continues to rise [1,2]. Globally, more than 30% of deaths annually (approximately 18.6 million) are due to cardiovascular diseases [3]. Although most cardiovascular events occur in older adults, young people with metabolic syndrome are at a high risk of future cardiovascular disease [4]. The risk factors for CVDs include obesity, hypertension, diabetes, and elevated serum lipid levels, and the atherosclerosis-related pathological process starts at a young age. Children with metabolic syndrome have biochemical and inflammatory factors that affect vascular physiology. Early assessment of metabolic syndrome in adolescents can help predict and prevent CVD risk later in life [5].

Assessment of metabolic syndrome in adolescents mainly includes body mass index (BMI), waist circumference (WC), blood pressure (BP), and levels of fasting glucose, triglycerides, and high-density lipoprotein cholesterol (HDL-c). Although no unified diagnostic criteria exist for metabolic syndrome among children and adolescents, the International Diabetes Federation (IDF) consensus definition and the definition proposed by de Ferranti et al. are the most widely used internationally [6,7,8]. The Taiwan Pediatric Association (TPA) also announced a set of diagnostic criteria for metabolic syndrome in children and adolescents in 2016, based on local age- and sex-specific BMI cutoff values. The TPA adopted BMI as a risk factor for metabolic syndrome due to a lack of age- and sex-specific WC values in Taiwan and lower clinical usage [9]. The global prevalence of metabolic syndrome in adolescents ranges from 0.3% to 26.4%, depending on the diagnostic criteria [10]. In Taiwan in 2003, the proportion of high school students with metabolic syndrome was 4.8% for boys and 3.9% for girls; in 2010–2011, the values were 4.12% and 3.01% among all adolescents when using TPA and IDF criteria, respectively [9,11].

Taiwan’s high school education is a three-year system, including senior high school and vocational high school, which can be divided into public and private systems. Different types of schools are associated with different familial socioeconomic backgrounds [12,13]. Students from higher socioeconomic status usually have higher academic achievement and attend public senior high schools, whereas those with lower socioeconomic status tend to attend private vocational schools [12,14,15,16]. Adolescents with lower socioeconomic status have poor academic performance and unhealthy habits, such as poor diet and smoking, thus increasing the risk of metabolic syndrome [17]. Although many factors influence the risk of metabolic syndrome in early childhood and adolescence, metabolic syndrome can be reversed by lifestyle modifications such as reducing weight, increasing physical exercise, and receiving adequate health education [18]. In Taiwan, senior and vocational high schools have different academic curricula, but the same curriculum for physical education. Many studies have investigated the prevalence of metabolic syndrome among adolescents, but few have analyzed differences in the prevalence among students attending different types of high schools. In this study, we investigated differences in the prevalence of metabolic syndrome between senior and vocational high school students; our findings can help guide the development of effective physical education curricula and health education programs for different types of high schools.

## 2. Materials and Methods

### 2.1. Study Population

The Taipei City government has offered health check-ups, including anthropometric measurements, physical examinations, and blood tests, for all first-year senior high and vocational high school students (grade 10) in Taipei City since 2011. These health examinations are administered by Taipei City Hospital every year from September to December (the first semester), soon after the students enter high school. Taipei City’s Department of Education requires the consent of every student’s parent(s) for the health check-ups and blood tests.

This cross-sectional study retrospectively collected anonymized data from all first-year high school students from Taipei City Hospital from 2011 to 2014. The data revealed that 46,182, 43,115, 40,241, and 36,595 students started senior and vocational high schools in Taipei City in 2011, 2012, 2013, and 2014, respectively. Each student’s BP, height, weight, WC, fasting blood glucose, total cholesterol, triglyceride, and HDL-c levels were recorded. We excluded students who were younger than 15 or older than 17 years old, did not fast for 8 h, had missing age or laboratory results, did not undergo the health examination, and had wrong or extreme height, weight, WC, or laboratory data. Ultimately, we included 81,076 and 68,863 first-year senior and vocational high school students, respectively (Figure 1). The study was approved by the Taipei City Hospital Research Ethics Committee (TCHIRB-1000908, TCHIRB-1010902, and TCHIRBB-10703112-E). All study procedures were conducted in accordance with the relevant guidelines and regulations of the Taipei City Hospital Research Ethics Committee.

### 2.2. Metabolic Syndrome and Its Risk Components

We used three definitions of metabolic syndrome in children and adolescents to investigate abnormal parameters and metabolic syndrome among students from different high schools: (1) (defined by the TPA, for children aged 8–18 years) central obesity (BMI > 95th percentile of sex- and age-specific groups) and the presence of two or more other clinical features (systolic BP (SBP) ≥ 130 mmHg or diastolic BP (DBP) ≥ 85 mmHg, triglycerides: ≥ 150 mg/dL, HDL-c: boys < 40 mg/dL and girls < 50 mg/dL, fasting blood glucose: ≥ 100 mg/dL, or currently under treatment); (2) (defined by the IDF consensus, children aged ≥ 10 years) central obesity (≥ 90th percentile of WC among children aged 10 to < 16 years, WC ≥ 90 cm in boys and ≥ 80 cm in girls among children aged ≥ 16 years) and the presence of two or more other clinical features (SBP ≥ 130 mmHg or DBP ≥ 85 mmHg, triglycerides ≥ 150 mg/dL, HDL-c < 40 mg/dL among children aged 10 to < 16 years, boys < 40 mg/dL and girls < 50 mg/dL among children aged ≥16 years, and fasting blood glucose: ≥100 mg/dL); and (3) (defined by de Ferranti et al.) the presence of three or more of the following five clinical features: central obesity (>75th percentile of WC), BP > 90th percentile, HDL-c: <50 mg/dL or <45 mg/dL among boys aged 15–19, triglycerides: ≥100 mg/dL, and fasting blood glucose: ≥110 mg/dL [6,7,9]. Although most definitions use WC as the indicator of central obesity, the TPA uses BMI because no WC percentiles exist in nationally representative samples of Taiwanese children and adolescents. Although WC is a better predictor of metabolic syndrome, both BMI and WC are useful screening tools for identifying metabolic syndrome among adolescents [19]. The definitions for each risk component of metabolic syndrome in adolescents according to TPA, IDF, and de Ferranti et al. are presented in Table 1.

### 2.3. Measurements

Anthropometric parameters (height, weight, and WC) were measured by well-trained technicians, with the students wearing a thin layer of clothes and no shoes. First, the technicians measured every student’s height and weight using a NAGATA P1200WH measuring machine. Next, the technicians asked the students to stand in a relaxed posture with arms at their sides, feet positioned close together, and weight evenly distributed across the feet and used the nonelastic tape to measure their WC at the end of their normal expiration. WC was measured approximately at the midpoint between the last palpable rib and the top of the iliac crest, following the WHO protocol [20]. After at least 10 min of rest, the students were asked to be seated, and nurses measured their SBP and DBP with an automatic sphygmomanometer (TERUMO ESP2000). Finally, students’ blood samples were drawn by clinical laboratory technologists using standard venipuncture technique. Fasting blood glucose, total cholesterol, triglycerides, and HDL-c were measured at the central laboratory of Taipei City Hospital. The details have been described previously [21].

### 2.4. Statistical Analysis

To determine eligibility for metabolic syndrome, we stratified the age into 15, 15.5, 16, 16.5 years; for example, students who were ≥ 15 and < 15.5 years were categorized as 15 years. The schools were defined as public, private, senior, or vocational high schools based on Taipei City’s Department of Education categorization. We calculated the means and standard deviations for continuous characteristics and proportions of binary characteristics. We used the *t*-test and chi-square test to determine the univariate association, and the significance level was set at ≤ 0.05. Multivariable regression was used to investigate factors associated with metabolic syndrome and different types of high schools. All statistical procedures were performed using SAS version 9.4 (SAS Institute, Cary, NC, USA).

## 3. Results

### 3.1. Characteristics of Participants

We included 149,939 high school students: 81,076 (54.07%) and 68,863 (45.93%) from senior and vocational high schools, respectively. Table 2 presents the general characteristics of all students and comparisons between senior and vocational high school students. Vocational high school students were significantly older than senior high school students. The proportion of the 16.5 age group was 0.75% (*n* = 612) and 2.55% (*n* = 1757) for senior and vocational high schools, respectively. Compared with senior high school students, vocational high school students had higher BMI, larger WC, higher FBG, higher TCHO levels, higher triglyceride levels, lower HDL-c, and higher SBP and DBP. Students in senior high schools were mainly in public schools, and most vocational high school students were in private schools. 

### 3.2. Prevalence of Metabolic Syndrome Risk Factors

Table 3 presents the number of risk factors that senior and vocational high school students had and the prevalence of metabolic syndrome among them, according to different diagnostic criteria. Only 67.42%, 72.78%, and 52.58% of high school students did not have any metabolic syndrome risk factors when applying the TPA, IDF, and de Ferranti et al. criteria, respectively. Vocational high school students had a higher proportion of metabolic syndrome risk factors, regardless of the criteria used. When the de Ferranti criteria were applied, only 55.26% of senior high school students and 49.43% of vocational high school students did not have any risk factors. In this study, the overall prevalence of metabolic syndrome was 1.73% (senior and vocational high school students: 1.22% and 2.33%, respectively) when applying the TPA criteria, 1.02% (0.69% and 1.40%, respectively) when using IDF criteria, and 5.11% (3.92% and 6.51%, respectively) when applying de Ferranti criteria. The most prevalent risk factors in both student groups were increased BP and central obesity.

Table 4 presents the associations of metabolic syndrome with the type of high school after adjustment for independent variables. After adjustment for other independent factors, boys had higher odds of having metabolic syndrome under all three criteria: TPA, aOR: 1.96 (95% confidence interval (95% CI): 1.80–2.13), R^2^: 0.026; IDF, aOR: 3.18 (95% CI: 2.82–3.58), R^2^: 0.040; de Ferranti, aOR: 2.07 (95% CI: 1.98–2.16), R^2^: 0.019. No difference was noted between different age groups in metabolic syndrome prevalence. Compared with students in public senior high schools, students in private senior high schools had 49% (95% CI: 1.28–1.72), 40% (95% CI: 1.15–1.71), and 31% (95% CI: 1.21–1.41) higher odds of having metabolic syndrome when applying the TPA, IDF, and de Ferranti criteria, respectively; students in public vocational high schools had 72% (95% CI: 1.54–1.93), 82% (95% CI: 1.58–2.11), and 53% (95% CI: 1.45–1.62) higher odds, respectively; and students in private vocational high schools had 2.30 (95% CI: 2.09–2.52), 2.33 (95% CI: 2.06–2.64), and 1.79 (95% CI: 1.71–1.88) higher odds, respectively.

## 4. Discussion

The prevalence of childhood and adolescent metabolic syndrome ranges from 3.98% to 8.4% when applying the IDF criteria, and is 8.91% when using the de Ferranti et al. criteria [22,23]. Among overweight and obese adolescents, the prevalence of metabolic syndrome is even higher, in the range of 24.09–56.32% [22]. The TPA criteria use BMI as a proxy for central obesity. The overall prevalence of metabolic syndrome among all high school students in this study was 1.73%, 1.02%, and 5.11% when applying the TPA, IDF, and de Ferranti criteria, respectively. We observed that more boys had metabolic syndrome than girls after adjustment for age and school type, consistent with previous studies [22,24]. Many other factors affect whether adolescents have metabolic syndrome, such as socioeconomic status, diet, exercise habits, sleep hours, and harmful habits such as smoking and drinking [17,25]. This study found that first-year students attending senior or vocational high school have differences in the prevalence of metabolic syndrome that may be related to the aforementioned factors; however, data on these risk factors were not available. Future studies on this topic should collect data on these factors to validate our results. Furthermore, the current lack of a consistent definition of metabolic syndrome in adolescents may be because of the unstable nature of adolescent metabolic syndrome and the lack of clarity regarding the predictive value of metabolic syndrome for future adult health. The latest study using a community-representative adolescent cohort in southern Taiwan indicated that the clustering structure of cardiovascular risk factors remained stable across adolescence during a two-year follow-up period. Further longitudinal follow-up research is required to clarify the risk of future cardiovascular diseases among adolescents with metabolic syndrome [26].

In the United States, the obesity rate among adolescents (12–19 years) displayed a nonsignificant but upward trend from 2003–2004 (17.4%) to 2011–2012 (20.5%) [27]. Obese children are more likely to have diabetes, hypertension, and dyslipidemia than healthy children [28]. Our study indicated that 7–24.77% of senior high school students and 11.23–29.74% of vocational high school students met the criteria for central obesity, and lifestyle interventions are recommended. Although central obesity did not increase from 2011 to 2014, more research is required to determine the long-term trends. A recent meta-analysis concluded that the prevalence of hypertension in children and adolescents worldwide was approximately 4%, with a peak at age 14, and more than 10% of students had high BP [29]. Between 2011 and 2012, the prevalence of dyslipidemia among the United States children and adolescents was approximately 20.2% [30]. Between 1996 and 2006, the proportion of dyslipidemia among junior high school students in Taiwan increased significantly—from 13% to 22.3% [31]. We found that more students in vocational high school had low HDL-c and elevated TG than those in senior high schools. The prevalence of impaired fasting glucose among the United States adolescents from 2005 to 2016 was 11.1% [32]. In our study, approximately 3% of students had impaired fasting glucose, with a higher prevalence among vocational high school students. Although the proportion may be lower than that found in previous studies, the differences in prevalence between high schools of different types should be given attention.

Similar to the secondary education system in many countries, in Taiwan, after three years of junior high school, students must take a high school entrance examination to decide which type of high school—senior or vocational, public or private—they will attend. Generally, students tend to choose public senior high schools first, especially those students with high academic achievement; the choice of the type of high school is rarely based on interest [33,34]. These schools have different curricular academic emphases and different professional course arrangements. However, for physical education and health education courses, they all follow the same recommendations from the educational supervisory unit [34,35]. Our study indicated that more students in vocational high schools had metabolic syndrome, regardless of the criteria used. Students in public and private vocational high schools had higher odds of having metabolic syndrome than public senior high school students, even after adjustment for age and sex. Thus, a single physical education and health education system might not meet the needs of all high school students in different schools.

Metabolic syndrome can be reversed through exercise and health education. Adolescents who ride bicycles to school have better body posture and a lower probability of metabolic syndrome than those who take transportation [36]. A meta-analysis indicated that a low level of physical activity results in metabolic syndrome [37]. Physical education can help increase physical activity to reduce metabolic syndrome in adolescents [38]. However, physical education in Taipei City is currently planned to consist of two sessions a week for high school students, and does not differ between the two types of high school. Our results imply that schools with a relatively high prevalence of metabolic syndrome, such as private vocational schools, should have more robust physical education curriculums. In addition, cardiovascular risk in adolescents who already have metabolic syndrome should be given attention, and these students should be provided with individualized physical education or enhanced health education courses.

Teenagers in Asia have fewer exercise hours and generally display fewer exercise habits. The World Health Organization recommends that adolescents engage in ≥60 min of moderate-to-vigorous physical activity (MVPA) per day. In the United States, 25.9% of high school students have physical education lessons every school day (all five weekdays), and both public and private high schools have an average of 3.3 days of physical education per week [39,40]. In Taiwan, the average MVPA of high school students is 19.7 min per day, much less than that recommended by the WHO [41]. This may be because in European and North American countries, school is usually dismissed in the afternoon, which is much earlier than in Asian countries. Early school dismissal allows students more exercise time. In addition, studies from Taiwan and Japan indicated that vocational high schools have a significantly higher proportion of students with low monthly household incomes [41,42,43]. Although students in Taiwan can exercise on holidays or after school, vocational high school students may need to work part-time after school to help their families [44]. Therefore, the physical education curriculum should be strengthened in high schools, particularly vocational high schools.

Few studies have investigated the prevalence of metabolic syndrome among adolescents in different high school settings. We used physical examination and laboratory data from all students enrolled in senior or vocational high schools from 2011 to 2014 in Taipei City. Our findings can serve as references for planning physical and health education courses for students in different types of high schools.

This study has a few limitations. First, although this study included all first-year high school students in Taipei City, the findings might not be generalizable to other cities or countries. Second, this was a cross-sectional study and did not collect information regarding the students’ family background and lifestyle behaviors. Many factors may cause metabolic syndrome in adolescents, including genetic and environmental factors, ethnic background, and lifestyle (physical activity, sleep pattern, smoking, alcohol use, and food intake) [24]. Moreover, the European HELENA study indicated that the probability of metabolic syndrome development is higher in adolescents with more socioeconomic disadvantages [45]. Because of data limitations, we could not determine the cause of the differences in the prevalence of metabolic syndrome between high school types. Future studies focusing on these possible associated factors are required. 

## 5. Conclusions

The prevalence of metabolic syndrome was significantly higher in vocational students than in senior high school students. Adolescent metabolic syndrome is a high-risk factor for future cardiovascular disease. Our findings emphasize the importance of revising the physical education curriculum in high schools to reduce interschool differences in metabolic syndrome risk. Furthermore, the amount of physical education designated in the curriculum should be increased and the health education program of schools should be strengthened to decrease the risk and prevalence of metabolic syndrome and promote adolescent health.

## Figures and Tables

**Figure 1 nutrients-14-03626-f001:**
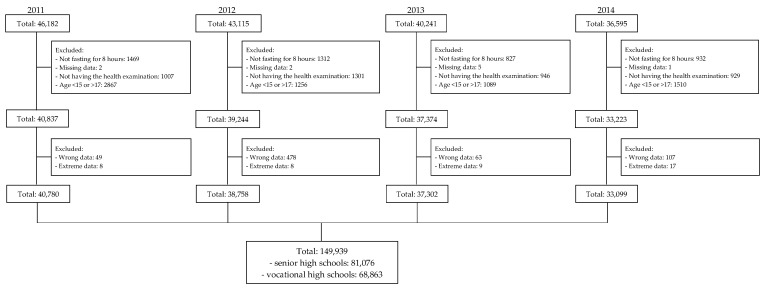
Flowchart of participant selection.

**Table 1 nutrients-14-03626-t001:** Definitions of metabolic syndrome in adolescents by TPA, IDF, and de Ferranti et al.

Required Criteria	Taiwan Pediatric Association (2016)	International Diabetes Federation (2007)	De Ferranti et al. (2004)
	Central Obesity + Two Out of Four *	Central Obesity + Two Out of Four *	Three Out of Five *
Central Obesity	BMI > 95th percentile of sex- and age-specific groups 15 (male/female): >25.4/25.2 15.5 (male/female): >25.5/25.3 16 (male/female): >25.6/25.3 16.5 (male/female): >25.6/25.3	10–<16 years of age: WC ≥ 90th percentile, ≥16 years of age: WC ≥ 90 cm (Asia male) WC ≥ 80 cm (Asia female)	WC > 75th percentile
Blood pressure	Systolic ≥ 130 mmHg or Diastolic ≥ 85 mmHg or >95th percentile for sex-age-specific groups	Systolic ≥ 130 mmHg or Diastolic ≥ 85 mmHg	>90th percentile
HDL-Cholesterol	<40 mg/dL (male) <50 mg/dL (female)	10–<16 years of age: < 40 mg/dL, ≥16 years of age: <40 mg/dL (male) <50 mg/dL (female)	<50 mg/dL boys aged 15–19: <45 mg/dL
Triglycerides	≥150 mg/dL	≥150 mg/dL	≥100 mg/dL
Fasting blood glucose	≥100 mg/dL or diagnosis of type 2 DM	≥100 mg/dL or diagnosis of type 2 DM	≥110 mg/dL

BMI: Body Mass Index (weight/height^2^; Kg/m^2^); WC: Waist Circumference; HDL: High-Density Lipoprotein; DM: Diabetes Mellitus; *: or under treatment.

**Table 2 nutrients-14-03626-t002:** Basic characteristics of senior and vocational high school students.

Variables	Senior High School (*n* = 81,076)	Vocational High School (*n* = 68,863)	*p-*Value
Sex			
Male	41,037 (50.62%)	36,089 (52.41%)	<0.001
Female	40,039 (49.38%)	32,774 (47.59%)	
Age	15.68 ± 0.31	15.69 ±0.35	<0.001
Age group			
15 (years)	25,623 (31.60%)	22,979 (33.37%)	<0.001
15.5 (years)	41,734 (51.48%)	32,651 (47.41%)	
16 (years)	13,107 (16.17%)	11,476 (16.66%)	
16.5 (years)	612 (0.75%)	1757 (2.55%)	
Height (cm)	165.35 ± 8.02	164.56 ± 8.17	<0.001
Weight (Kg)	57.86 ± 12.23	58.92 ± 14.06	<0.001
BMI (Kg/m^2^)	21.07 ± 3.63	21.56 ± 4.35	<0.001
WC (cm)	69.22 ± 9.33	70.30 ± 10.94	<0.001
FBG (mg/dL)	83.88 ± 8.56	85.37 ± 10.76	<0.001
TCHO (mg/dL)	159.80 ± 27.97	160.80 ± 28.39	<0.001
TG (mg/dL)	68.86 ±29.08	72.10 ± 32.94	<0.001
HDL-c (mg/dL)	61.11 ± 12.64	59.60 ± 12.84	<0.001
SBP (mmHg)	113.10 ± 15.09	114.00 ± 15.88	<0.001
DBP (mmHg)	63.26 ± 10.31	63.88 ± 10.91	<0.001
School-Type			
Public school	66,587 (82.13%)	24,103 (35.00%)	<0.001
Private school	14,489 (17.87%)	44,760 (65.00%)	

BMI: body mass index; WC: waist circumference; FBG: fasting blood glucose; TCHO: total cholesterol; TG: triglyceride; HDL-c: high-density lipoprotein cholesterol; SBP: systolic blood pressure; DBP: diastolic blood pressure.

**Table 3 nutrients-14-03626-t003:** Distribution and prevalence of metabolic syndrome risk factors for senior and vocational high school students.

	Taiwan Pediatric Association			International Diabetes Federation			De Ferranti et al.		
Numbers of Metabolic Syndrome Risk Factors	Total (*n* = 149,939)	Senior High School (N = 81,076)	Vocational High School (N = 68,863)	Total (*n* = 149,939)	Senior High School (N = 81,076)	Vocational High School (N = 68,863)	Total (*n* = 149,939)	Senior High School (N = 81,076)	Vocational High School (N = 68,863)
		*n* (%)	*n* (%)		*n* (%)	*n* (%)		*n* (%)	*n* (%)
0	101,096 (67.42)	56,977 (70.28)	44,119 (64.07)	109,133 (72.78)	61,247 (75.54)	47,886 (69.54)	78,841 (52.58)	44,800 (55.26)	34,041 (49.43)
1	35,834 (23.90)	18,565 (22.90)	17,269 (25.08)	32,115 (21.42)	16,250 (20.04)	15,865 (23.04)	45,314 (30.22)	24,386 (30.08)	20,928 (30.39)
2	10,301 (6.87)	4502 (5.55)	5799 (8.42)	7003 (4.67)	2955 (3.64)	4048 (5.88)	18,117 (12.08)	8708 (10.74)	9409 (13.66)
3	2318 (1.55)	909 (1.12)	1409 (2.05)	1434 (0.96)	543 (0.67)	891 (1.29)	6252 (4.17)	2644 (3.26)	3608 (5.24)
4	374 (0.25)	120 (0.15)	254 (0.37)	243 (0.16)	78 (0.10)	165 (0.24)	1364 (0.91)	530 (0.65)	834 (1.21)
5	16 (0.01)	3 (< 0.01)	13 (0.02)	11 (0.01)	3 (< 0.01)	8 (0.01)	51 (0.03)	8 (0.01)	43 (0.06)
Prevalence									
Metabolic syndrome	2593 (1.73)	988 (1.22)	1605 (2.33)	1528 (1.02)	563 (0.69)	965 (1.40)	7667 (5.11)	3182 (3.92)	4485 (6.51)
Central Obesity	20,671 (13.79)	9136 (11.27)	11,535 (16.75)	13,412 (8.94)	5679 (7.00)	7733 (11.23)	40,563 (27.05)	20,083 (24.77)	20,480 (29.74)
Increased BP	24,093 (16.07)	12,224 (15.08)	11,869 (17.24)	24,093 (16.07)	12,224 (15.08)	11,869 (17.24)	26,846 (17.90)	13,440 (16.58)	13,406 (19.47)
Low HDL-c	11,742 (7.83)	5676 (7.00)	6066 (8.81)	5485 (3.66)	2458 (3.03)	3027 (4.40)	18,395 (12.27)	8741 (10.78)	9654 (14.02)
Elevated TG	3865 (2.58)	1690 (2.08)	2175 (3.16)	3865 (2.58)	1690 (2.08)	2175 (3.16)	19,354 (12.91)	9286 (11.45)	10,068 (14.62)
High FBG	4595 (3.06)	2065 (2.55)	2530 (3.67)	4595 (3.06)	2065 (2.55)	2530 (3.67)	857 (0.57)	344 (0.42)	513 (0.74)

BP: blood pressure; HDL-c: high-density lipoprotein cholesterol; TG: triglycerides; FBG: fasting blood glucose.

**Table 4 nutrients-14-03626-t004:** Adjusted odds ratios of adolescents having metabolic syndrome as defined by different criteria.

		Taiwan Pediatric Association				International Diabetes Federation				De Ferranti et al.			
Variables		aOR	95% CI		*p*-Value	aOR	95% CI		*p*-Value	aOR	95% CI		*p*-Value
Sex	Female	reference			-	reference			-	reference			-
	Male	1.96	1.80	2.13	< 0.001	3.18	2.82	3.58	<0.001	1.49	1.42	1.56	<0.001
Age group	15	reference			-	reference			-	reference			-
	15.5	0.93	0.85	1.01	0.09	0.96	0.86	1.08	0.54	0.92	0.87	0.97	0.01
	16	0.91	0.81	1.02	0.12	1.01	0.87	1.17	0.91	0.93	0.87	0.99	0.04
	16.5	0.83	0.62	1.12	0.22	1.05	0.74	1.48	0.80	0.89	0.74	1.06	0.19
high schools	public senior	reference			-	reference			-	Reference			-
	private senior	1.49	1.28	1.72	< 0.001	1.40	1.15	1.71	0.001	1.31	1.21	1.43	<0.001
	public vocational	1.72	1.54	1.93	< 0.001	1.82	1.58	2.11	<0.001	1.59	1.48	1.69	<0.001
	private vocational	2.30	2.09	2.52	< 0.001	2.33	2.06	2.64	<0.001	1.91	1.80	2.01	<0.001

## Data Availability

Data are available upon reasonable request from the corresponding author.

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
