# Peer review of "Metabolic Syndrome Prevalence among High School First-Year Students: A Cross-Sectional Study in Taiwan"

_nutrients, 2022, doi:10.3390/nu14173626_

Round 1

Reviewer 1 Report

Thank you for the opportunity to review your manuscript. I clearly note the high cognitive value of the work. Nevertheless, I would ask the Authors to make a few corrections:

1. perhaps in the title do not emphasize that it is about the 'senior' and 'vocational' levels, as at the first moment for a foreign reader the topic may be confusing. I suggest leaving 'high school first-year students' alone.

2. please demonstrate the hypotheses implemented within the work.

3. the authors calculated BMI for the students, and as you know, this is an indicator for people over the age of 18. Its use is possible when certain conditions are met, please describe them in detail in the text.

4. please pluralize the dots before the citation links [ ].

5. in tables, please replace commas ( , ) with periods ( . ).

Greetings!

Reviewer 2 Report

This study used 3 definitions to examine the prevalence of MetS between senior high school and vocational school students. The MetS rates, regardless of definition used, were much higher among vocational high school students. The findings were interesting and provided some implications for future research and clinical practice, as well as public health interventions.

1. Intro:

The authors should provide a bit more contexts on why it was necessary to compare the students from two different types of schools. Did they have different socioeconomic standings? Other factors that might cause differences in cardiovascular outcomes? Some justification would be good to provide the readers with some contexts. The differences in student body of private vs. public schools should also be mentioned. Some of this information was discussed in the Discussion section, but should also be mentioned earlier to provide contexts and justification.

Methods

1. It was hard to compare the three different definitions of MetS in writing. The authors might want to consider making a table to better outline the similarities and differences in these three definitions.

2. What was the sampling strategy? Convenience sampling? Should be mentioned.

3. Did the survey measure any other modifiable lifestyle behaviors such as alcohol use, tobacco use, and sleep pattern? All three behaviors have been linked to MetS. If not, this should be mentioned in the discussion section. It was possible that the differences in MetS rates across different types of schools were due to these behaviors.

Results

1. For the multivariate regression results, the goodness-of-fit results of the models should be presented in the table, along with the adjusted OR.

Round 2

Reviewer 2 Report

The authors made sufficient changes to the manuscript. I recommend that this manuscript should be accepted for publication.